# Research Progress on Taxus Extraction and Formulation Preparation Technologies

**DOI:** 10.3390/molecules29102291

**Published:** 2024-05-13

**Authors:** Xinyu Gao, Kuilin Chen, Weidong Xie

**Affiliations:** 1State Key Laboratory of Chemical Oncogenomics, Shenzhen International Graduate School, Tsinghua University, Shenzhen 518055, China; gao-xy23@mails.tsinghua.edu.cn (X.G.); ckl23@mails.tsinghua.edu.cn (K.C.); 2Shenzhen Key Laboratory of Health Science and Technology, Institute of Biopharmaceutical and Health, Shenzhen International Graduate School, Tsinghua University, Shenzhen 518055, China

**Keywords:** *Taxus* extracts, preparation methods, pharmaceutical design, drug preparation

## Abstract

*Taxus*, as a globally prevalent evergreen tree, contains a wealth of bioactive components that play a crucial role in the pharmaceutical field. *Taxus* extracts, defined as a collection of one or more bioactive compounds extracted from the genus *Taxus* spp., have become a significant focus of modern cancer treatment research. This review article aims to delve into the scientific background of *Taxus* extracts and their considerable value in pharmaceutical research. It meticulously sifts through and compares various advanced extraction techniques such as supercritical extraction, ultrasound extraction, microwave-assisted extraction, solid-phase extraction, high-pressure pulsed electric field extraction, and enzymatic extraction, assessing each technology’s advantages and limitations across dimensions such as extraction efficiency, extraction purity, economic cost, operational time, and environmental impact, with comprehensive analysis results presented in table form. In the area of drug formulation design, this paper systematically discusses the development strategies for solid, liquid, and semi-solid dosage forms based on the unique physicochemical properties of *Taxus* extracts, their intended medical uses, and specific release characteristics, delving deeply into the selection of excipients and the critical technical issues in the drug preparation process. Moreover, the article looks forward to the potential directions of *Taxus* extracts in future research and medical applications, emphasizing the urgency and importance of continuously optimizing extraction methods and formulation design to enhance treatment efficacy, reduce production costs, and decrease environmental burdens. It provides a comprehensive set of preparation techniques and formulation optimization schemes for researchers in cancer treatment and other medical fields, promoting the application and development of *Taxus* extracts in pharmaceutical sciences.

## 1. Introduction

*Taxus*, a genus within the Taxaceae family, is a precious evergreen conifer distributed across various regions globally. Species such as *Taxus baccata* Thunb, *Taxus cuspidate* Siebold & Zucc, *Taxus chinensis* (Pilg.) Rehder, *Taxus mairei* (Lemée & H. Lév.) S. Y. Hu ex T. S. Liu and *Taxus wallichiana* Zucc. are prevalent [1]. Its bioactive compounds possess significant pharmacological effects [2,3,4], rendering *Taxus* extract a valuable resource in modern drug development and sparking considerable scientific interest in its in-depth study. With the advancement of pharmaceutical sciences, developing effective extraction and application strategies to utilize the medicinal value of *Taxus* fully has become a research focal point.

In the evolution of extraction technologies, supercritical CO_2_ extraction, ultrasound extraction, aqueous extraction, chemical extraction, and enzymatic catalysis have been widely applied to extract active components from *Taxus*. Each method showcases unique advantages and application conditions. For example, supercritical CO_2_ extraction stands out for its environmental friendliness and high efficiency, while aqueous and chemical extraction methods are broadly adopted for their simplicity and low cost. These diverse extraction technologies not only facilitate the effective acquisition of *Taxus*’s active ingredients but also lay a solid foundation for subsequent drug development and application.

As extraction processes continue to be optimized and developed, the formulation design of *Taxus* extracts becomes a critical step in realizing their potential medical applications. Precise control over the dosage form can significantly enhance drug bioavailability, extend the duration of action, and reduce side effects, thereby achieving more accurate and effective therapeutic effects in areas such as cancer treatment. Solid, liquid, and semi-solid dosage forms are developed and optimized based on their physicochemical properties and anticipated clinical applications to meet various therapeutic needs.

This article reviews the latest advancements in *Taxus* extracts’ preparation methods and formulation design. By comparing and analyzing the efficiency, cost, time, and environmental impact of different extraction techniques, it aims to provide comprehensive scientific guidance and reference for applying and developing *Taxus* extracts. Moreover, by exploring the principles and strategies of formulation design, this article further elucidates how to enhance the medicinal application value of *Taxus* extracts through dosage form optimization, offering essential insights and methods for future research directions and medical practice.

## 2. Preparation Methods of *Taxus* Extracts

Efficient extraction of active components from medicinal herbs is crucial for studying their pharmacological effects and mechanisms. *Taxus* extracts, rich in taxane, flavonoids, polysaccharides, steroids, and alkaloids, represent a diverse composition of bioactive compounds [2,3,4]. Currently, many preparation methods exist for *Taxus* extracts, with researchers focusing not only on improving extraction efficiency and cost-effectiveness but also on the sustainability impact on the environment. In the following sections, we will detail various extraction technologies, each with unique advantages and limitations, showcasing their suitability in different application contexts. Ultimately, we aim to conduct an in-depth comparison and analysis of these methods to identify the most optimized *Taxus* extraction technology, striving to ensure high-quality extracts while considering cost-effectiveness and environmental impact.

### 2.1. Supercritical CO_2_ Extraction

#### 2.1.1. Characteristic

Supercritical Fluid Extraction (SFE) leverages the distinct physicochemical properties of supercritical fluids, widely used in various industries, including pharmaceutical [5,6,7]. Supercritical Fluid Extraction (SFE) uses the unique physicochemical properties of supercritical fluids, with carbon dioxide (CO_2_) being the most commonly employed due to its favorable properties and extraction efficiency. CO_2_ is non-toxic, inert, non-flammable, inexpensive, recyclable, and environmentally benign, which makes it an ideal solvent. Its critical temperature of 31.1°C is particularly suitable for extracting thermolabile substances, allowing processes to occur at temperatures close to ambient, thereby preserving the integrity of heat-sensitive biological compounds [8]. Focused on Taxus, supercritical CO_2_ extraction is renowned for its high selectivity, low-temperature operations preserving thermosensitive substances, and environmental friendliness [9]. These features, coupled with precise control over extraction parameters such as pressure and temperature, make SFE ideal for extracting delicate compounds like taxanes. The extraction mechanism involves CO_2_ penetrating the plant material, where it dissolves the target compound based on differential solubility in the supercritical phase. This process can be applied directly to natural products that often require some form of pretreatment such as drying, crushing, or grinding to increase the efficiency of the extraction [10,11].

#### 2.1.2. Compounds and Parameters of Supercritical CO_2_ Extraction from *Taxus*

Taxanes are key compounds in Taxus, valued for their therapeutic properties. Supercritical CO_2_ extraction excels in efficiency and selectivity, especially when paired with chromatographic techniques. This method has been optimized to enhance the yield and purity of taxanes and other compounds, such as volatile oils and flavonoids. Specific parameters and outcomes for the extraction of various compounds are detailed in Table 1, illustrating the method’s broad applicability and effectiveness.

### 2.2. Maceration and Reflux Extraction

#### 2.2.1. Characteristic

Solvent extraction, a traditional method in plant-based compound extraction, utilizes the solubility differences of components in various solvents. This method is distinguished by two primary techniques: maceration and reflux extraction, which involve solvents that dissolve target components effectively while excluding non-target ones, enabling a selective and efficient separation process. Maceration, suitable for thermosensitive or volatile compounds, preserves the integrity and activity of these substances at low or room temperatures. It involves soaking the plant material in solvents such as ethanol, methanol, or ether, followed by separation of the extract through filtration or centrifugation. Reflux extraction, on the other hand, enhances the extraction efficiency by maintaining the solvent at boiling temperatures, which is particularly effective for extracting non-volatile, heat-stable components like total flavonoids [19]. The process includes placing plant material in a solvent, heating it to boiling, and condensing the vapors back to liquid to be reused, thus ensuring thorough extraction through continuous solvent circulation [20].

#### 2.2.2. Examples of Liquid Extractions from Taxus

This section discusses various applications of liquid solvent extraction methods, particularly focusing on their use in extracting valuable compounds from Taxus species. Each method’s adaptability to both cold and hot extraction processes caters to the stability requirements of diverse compounds:

Solvent extraction leverages the differential solubility of compounds in various solvents to isolate bioactive components from plant material, making it a cornerstone of plant-based compound extraction. This method is particularly effective for separating target compounds from non-target components and is widely used due to its adaptability to both cold and hot extraction processes, catering to the stability requirements of diverse compounds.

Cold Extraction is ideal for thermosensitive or volatile compounds, preserving the integrity and activity of these substances at low or room temperatures. The process involves soaking the plant material in solvents such as ethanol, methanol, or ether, followed by separation of the extract through filtration or centrifugation [21]. 

Hot Water Extraction employs water as a solvent for extracting water-soluble components like polysaccharides, leveraging water’s polarity to dissolve target compounds effectively [22]. This method often requires multiple steps of boiling and concentrating to optimize the yield and purity of the extracts [23].

Reflux Extraction, an enhanced form of hot extraction, maintains the solvent at boiling temperatures to improve the extraction efficiency of specific components such as total flavonoids [24]. The extract can also be prepared into freeze-dried powder for use, specifically by collecting the liquid from three extractions, filtering it with a vacuum filter, concentrating it in a water bath at 80 °C using a rotary evaporator, freeze-drying the concentrate into powder, and storing it at −80 °C [25]. 

For components with poor water solubility like taxanes, Organic Solvent Extraction is necessary. Specific parameters for extracting target compounds from *Taxus* with different solvents are shown in Table 2 below.

Traditionally, this involves solvents such as ethanol, methanol, and particularly dichloromethane (DCM), which, despite their effectiveness, are being phased out due to environmental and health risks [31,32]. There is a trend towards using safer and more sustainable solvents such as ethyl acetate and supercritical CO_2_ [33].

Modern advancements such as Accelerated Solvent Extraction (ASE) and Ultrasound-Assisted Extraction are designed to enhance extraction efficiency and reduce the environmental impact [34]. ASE uses mechanical or thermal energy to accelerate the extraction process, achieving high purity and yield in reduced time [26]. 

Innovative Approaches include techniques like anti-solvent recrystallization, which refines the purity of extracted compounds by precipitating the desired compound from solution, enhancing selectivity and efficiency. Despite the effectiveness of water-based methods, the extraction of taxane compounds often requires the use of organic solvents due to the compounds’ hydrophobic nature. Research has shown that using a co-solvent based on natural menthol improves extraction efficiency and reduces environmental impact. The ratio of menthol to isopropanol was 1:1 (mol/mol), with a solid-liquid ratio of 1:30 g/mL, extraction time of 30 min, ultrasound power of 250 W, and water content of 80%. Under these conditions, the total extraction efficiency of seven major taxanes from Chinese yew (*Taxus chinensis*) needles was 1.25 to 1.44 times higher than that of conventional solvent extraction methods [30]. The high cost of menthol limits its feasibility for industrial-scale applications. Furthermore, this method employs ultrasound-assisted extraction rather than relying solely on chemical extraction, increasing operational complexity. Therefore, while this co-solvent extraction method based on natural menthol demonstrates excellent efficiency and environmental friendliness, its high cost and operational complexity are major barriers to industrial application.

In recent years, despite the traditional nature of solvent extraction, there have been some innovations in solvent extraction methods. The principle of anti-solvent recrystallization is based on the effect of a solvent and anti-solvent mixture system on the solubility of the solute. This method typically involves dissolving the target substance in a solvent and then adding an anti-solvent that mixes well with the solvent but has a lower solubility for the target substance. By controlling the amount of anti-solvent added, the solubility of the target substance in the solution is reduced, prompting it to recrystallize [35,36]. Anti-solvent recrystallization can be used to separate and purify taxanes from crude extracts of *T. mairei*, with the best purification effect achieved using methanol as the solvent and water as the anti-solvent.

### 2.3. Ultrasound-Assisted Extraction

Ultrasound-assisted extraction (UAE) leverages the cavitation effect induced by ultrasound waves in a liquid medium to efficiently extract bioactive components from plant materials, including various *Taxus* species. The generation of microscopic bubbles, which upon bursting create high temperatures and pressures, disrupts plant cell structures, facilitating solvent penetration and component extraction [37]. This method is characterized by its simplicity, high efficiency, low operational temperatures, and reduced solvent consumption compared to traditional extraction methods like SFE and thermal reflux extraction [38].

In practice, UAE has been effective in extracting various compounds from Taxus. It often incorporates solvents like ethanol, which has been identified as particularly effective for extracting taxane compounds. The synergy between ultrasound waves and solvents enhances the extraction process, improving efficiency and yield [39]. The optimal extraction parameters for *Taxus wallichiana* var. mairei leaves include a material-liquid ratio of 1:15 (g/mL), extraction time of 23 min, and ultrasound temperature of 40 °C. Under these conditions, the content of 10-DAB was higher at 1.18%, compared to 0.546‰ by SFE, demonstrating the advantage of ultrasound extraction in enhancing the efficiency of specific component extraction [40]. The best extraction conditions for *Taxus cuspidata* were a liquid-material ratio of 53.23 mL/g, ultrasound time of 1.11 h, and ultrasound power of 207.88 W. Under these parameters, the average content of paclitaxel extracted was about 130.576 µg/g, significantly superior to other extraction methods [41]. Ionic liquids (ILs), considered “green” solvents, have been widely used in extraction and separation fields in recent years. They can be used to extract paclitaxel from *Taxus x media* using a methanol solution. The unique advantage of the combined application of ultrasound and ionic liquids in enhancing extraction efficiency is shown. Compared to traditional solvent extraction, UAE with methanol and magnetic ionic liquid (MIL) as an auxiliary significantly improves extraction yield, reduces the use of methanol, and shortens extraction time. Under optimized conditions of 1.2% IL, a 1:10.5 solid-liquid ratio, and 30 min of ultrasound irradiation, the extraction yield reached 0.224 mg/g [42].

Further integration of UAE with enzyme technology allows for greater selectivity and efficiency. Enzymatic breakdown of plant cell walls prior to ultrasound extraction improves the release and yield of specific active components [43]. For instance, the optimal conditions for extracting proanthocyanidins from Taxus mairei include a 60% ethanol solvent, an enzyme concentration of 0.15 g/L, and an ultrasound power of 120 W, resulting in a yield of 3.84% (extraction rate of 83.48%) [44].

In addition, in order to ensure the high efficiency and safety of biopharmaceuticals, the natural active compounds used as raw materials for development are usually required to have a high purity. To ensure the effectiveness of Taxanes in drug development, Zhang et al., used UAE and Preparative High-Performance Liquid Chromatography to extract 10-DAT and Paclitaxol from *Taxus cuspidata*. The optimal extraction rate was obtained with the liquid-to-solid ratio of 20.88 times, ultrasound power of 140.00 W, ultrasound time of 47.63 min, ethanol content of solvent of 83.50%, liquid chromatography flow rate of 10 mL/min, sample size of 0.5 mL and column temperature of 30 °C [45,46]. The purities of 10-DAT and Paclitaxol reached 95.33% and 99.15%, respectively. At the same time, Zhang et al., also used UAE and Antisolvent Recrystallization method to extract and purify taxane compounds from *Taxus cuspidata*. When the volume ratio of antisolvent to solvent was 28.16 times, the deposition temperature was 22.91 °C, and the deposition time was 1.76 min, the extraction purity of taxane compounds increased from 0.20% to 23.24% [47]. Moreover, the structure of the product is not destroyed, and the extraction efficiency is guaranteed while the activity is strong.

By comparing UAE and enzyme technology, we can see that both have their advantages in extracting active plant components. UAE is known for its high efficiency and low energy consumption, while enzyme technology is highlighted by its high selectivity and strong targeting. Combining these two technologies not only improves the extraction efficiency of specific components but also optimizes the extraction process through different mechanisms, achieving higher yields and better quality of extraction. The application of this integrated method provides a new perspective and possibility for the extraction of active plant components. For a detailed summary of the parameters used in ultrasound-assisted extraction and their respective yields across different Taxus species, please refer to Table 3.

### 2.4. Microwave-Assisted Extraction 

Microwave-Assisted Extraction (MAE) is a technique that uses microwave energy to accelerate the mass transfer process between the solvent and the sample for compound extraction. Its principle is based on the unique properties of microwave radiation, which can penetrate materials and accelerate molecular vibration, thereby generating heat [49,50]. Since microwave heating occurs from inside the sample, the heating process is relatively uniform, lacking the temperature gradient of traditional heating methods. Different substances have varying capacities for microwave absorption, allowing MAE to be optimized for specific compounds [51,52].

In practical applications, MAE has demonstrated significant advantages over conventional solvent extraction methods, including substantial reductions in both extraction time and solvent usage [53]. For example, optimal MAE conditions for Taxus baccata L. involve using a 90% methanol solution, achieving effective paclitaxel extraction within just 7 minutes compared to hours with traditional methods [54]. 

Ultrasound Microwave Synergistic Extraction (UME) combines the cavitation effect of ultrasound with the enhanced heat transfer (ionic conduction and molecular dipole rotation) effect of microwaves, accelerating the release of compounds inside cells for effective component extraction. The optimal conditions for taxane extraction were an ultrasound power of 300 W, microwave power of 215 W, temperature of 50 °C and a mesh size of 130. Under these conditions, the yield of taxanes was 570.32 μg/g, which is an increase of 13.41% and 41.63% compared to ultrasound (US) and microwave (MW) treatment, respectively [55].

Focusing on the marc left after paclitaxel extraction from *T. mairei* leaves, ultrasound-assisted water extraction-alcohol precipitation was used to extract polysaccharides, with glucose as the control and using the phenol-sulfuric acid method to determine polysaccharide content. Single-factor experiments were used to select the main factors affecting extraction, with an orthogonal experiment to optimize the extraction process finally. The optimal extraction conditions for polysaccharides from the marc of *T. mairei* leaves were a solid-liquid ratio of 1:30, ultrasound power of 50W, and ultrasound time of 20 min. Compared to the extraction rate of polysaccharides from *T. mairei*, the yield decreased from 5.12% to 3.73%, but the purity increased from 70.53% to 89.53%. The highest yield of polysaccharides was achieved at an alcohol precipitation concentration of 80% [56].

### 2.5. Solid-Phase Extraction

In the separation and enrichment of taxane compounds, Solid-Phase Extraction (SPE) technology, particularly using macroporous resin adsorption, has proven effective. Utilizing macroporous resins like AB-8, which excel in chemical adsorption, significantly enhances the specificity and efficiency of compound isolation [57]. Studies have shown that AB-8 resin outperforms other resins in adsorption capacity, optimizing extraction parameters such as solvent concentration and flow rate to improve yields dramatically. For instance, after treatment with AB-8 resin, the content of compounds like 7-xyl-10-DAT and 10-DAB increased significantly, with corresponding recovery rates of over 85% [58].

Furthermore, combining SPE with advanced adsorption technologies has simplified the extraction and purification processes compared to traditional methods. For example, using Diaion® HP-20 and silica-based SPE, this approach has substantially increased the yield of targeted compounds such as paclitaxel and 10-DAB, enhancing both efficiency and cost-effectiveness [59].

### 2.6. High-Intensity Pulsed Electric Field Extraction

High-Intensity Pulsed Electric Fields (PEF) is a new technique for extracting bioactive components from food and other natural products. The principle of PEF treatment is mainly based on the effect of the electric field on the cell membrane. When cells are exposed to a high-intensity electric field for a short duration, micropores form on the cell membrane, a phenomenon known as electroporation. Electroporation can be reversible or irreversible, depending on the intensity and duration of the electric pulse. Reversible electroporation allows for substance exchange without causing cell death, while irreversible electroporation leads to cell death, thereby releasing the cell’s internal components [60]. By adjusting PEF parameters such as electric field strength, pulse width, and number of pulses, the extraction efficiency of specific target components can be optimized [61]. This method can disrupt cell structures at lower temperatures, helping to maintain the integrity and activity of bioactive components [62]. A study extracting taxanes from *Taxus* explored the impact of seven extraction conditions on the yield of target compounds. Under conditions of an electric field strength of 16 kV/cm, pulse number of 8, particle size of 160 mesh, solid-liquid ratio of 1:60, single extraction, centrifugal speed of 8000 r/min, and flow rate of 7 mL/min, the maximum extraction yield reached 672.13 μg/g, 1.07–1.84 times that of the control, solid-liquid extraction (SL), and ultrasound extraction (US) groups [63].

### 2.7. Enzyme-Assisted Extraction

In recent years, enzymatic degradation technology has rapidly developed in the extraction process of traditional Chinese medicine due to its green and environmentally friendly nature, which is suitable for low temperatures. Using the bark of *Taxus yunnanensis* W.C.Cheng and L.K.Fu as raw material, the cellulase method can assist in extracting taxane compounds. In the experiment, after removing lipids and some liposoluble pigments with petroleum ether, drying the powder, adding cellulase solution, anhydrous ethanol, and assisted by ultrasound extraction, the results showed that the optimal process conditions are: enzyme treatment time of 1.0 h, enzyme dosage of 0.6%, enzyme treatment temperature of 45 °C, and solid-liquid ratio of 1:16 (g:mL). Under these conditions, the yield of taxane compounds was 0.898% [47], which is relative to no enzyme treatment. This extraction process is simple, with mild and stable extraction conditions, though the steps are relatively cumbersome.

An innovative method involves using a Low-Temperature, High-Efficiency Enzyme-Ultrasound Assisted Coupling Extraction (EUCE) method. Targeting anti-diabetic and anti-tumor activities, the extraction process of polysaccharides from the branches and leaves of *Taxus cuspidata* Siebold and Zucc. was optimized. The optimized conditions are: extraction temperature of 51 °C, extraction time of 33 min, solid-liquid ratio of 1:19 (g:mL), enzyme concentration of 0.10 mg/mL, with a polysaccharide yield of 4.78% ± 0.18% [64]. The extracted *Taxus* polysaccharides have significant biological activities, including inhibiting α-glucosidase and anti-tumor effects on various cancer cells, indicating the effectiveness of this method and the potential for extracting bioactive compounds from *Taxus*.

### 2.8. Comparison and Analysis of Different Extraction Methods

Each extraction method has its pros and cons, as summarized in Table 4 below:

## 3. Development of Dosage Forms for *Taxus* Extracts

Traditional Chinese medicine, a valuable resource in China’s medical field, primarily employs decoction methods, which, compared to Western medicine, have disadvantages such as large dosing, poor uniformity, and low therapeutic efficiency, severely affecting the development and application of traditional Chinese medicine formulations. *Taxus*, as a cherished medicinal plant resource in China, besides its application in the extraction of paclitaxel, is recorded in pharmacopeias such as “Chinese Herbal Medicine” and “Chinese Materia Medica” for its diuretic and swelling reduction effects through decoction. In recent years, with the advancement of active component separation technology and pharmacological activity research of *Taxus* extracts, active monomers from *Taxus* have been gradually isolated and identified. Further developing and preparing extracts into different types of pharmaceutical formulations is of great significance for improving the utilization rate of active components of *Taxus* and promoting the development of the *Taxus* industry.

As of now, for *Taxus* and its active components, the National Medical Products Administration (NMPA) and the “Traditional Chinese Medicine Formula Database” have included a total of 88 types of traditional Chinese medicines and domestic drugs, mainly in forms of injections and oral capsules. A search of currently registered *Taxus* drugs shows that the drug forms mainly include enteric-coated tablets, capsules, and tablets. Additionally, a search for *Taxus* medical patents resulted in 1434 entries, with drug forms including sustained-release agents, suspensions, emulsions, and other solid/semi-solid or liquid dosage forms. For instance, a Chinese patent CN114306290A discloses a respiratory method for anti-tumor drugs and their preparation methods, inhaling in gas form to circulate throughout the body for treating patients. Another Chinese patent CN101254217B discloses the application of *Taxus* extract in preparing oral anticancer drugs, extracting active taxane compounds from *Taxus*, and developing anticancer drug formulations such as droplets, suspensions, and emulsions, effectively improving the low oral bioavailability of paclitaxel and reducing adverse reactions of traditional paclitaxel injections to some extent. This review of current research progress on *Taxus* formulations aims to provide adequate references for subsequent formulation development.

### 3.1. Solid Dosage Forms

Many solid formulations such as capsules and tablets have been developed from *Taxus* extracts in recent years. Besides capsule and tablet dosage forms, granule forms [66,67] are also worth developing. Scholars in the field of *Taxus* research focus on its anticancer properties, but it also has various other pharmacological activities, such as antibacterial and anti-inflammatory effects. Compound enteric-coated capsules containing *Taxus* extracts have been clinically used for adjuvant treatment of lung cancer. According to data from the China Clinical Trial Registry (ChiCTR), years of clinical application have shown the effectiveness and safety of compound *Taxus* capsules in treating various tumors such as lung, breast, stomach, esophageal, and ovarian cancer. Additionally, clinical trials on compound *Taxus* capsules for treating non-small cell lung cancer have also commenced, indicating that the development of existing *Taxus* drugs still focuses on cancer treatment. Additionally, studies have shown that *Taxus* extracts have specific activities in antibacterial and antifungal aspects [68,69]. Ghaedi et al. combined *Taxus* extracts with silver-based compounds (Ag NPs) to form solid tablets, which showed strong inhibitory activity against *E. coli*, *Staphylococcus aureus*, and the fungus *Aspergillus*, while maintaining high stability [70].

Traditional *Taxus* extracts’ active components often struggle to reach target sites in the body due to low bioavailability, hindering their anticancer properties. Current research indicates that the bioavailability of active components in *Taxus* extracts, such as 10-DAB, 7-epi-10-DAT, 7-epi-paclitaxel, and paclitaxel, only reaches 7.3–37% when used in conjunction with P-gp inhibitors and other drugs [66,67,71]. Due to low bioavailability, key active components of *Taxus*, such as paclitaxel drugs, are often administered as injections via intravenous routes to bypass the intestinal barrier and exert their therapeutic effects. However, studies have shown that specific nanoparticle colloidal dispersion systems can promote drug absorption and passive targeting to some extent [72]. Kajani et al. used *Taxus* extracts as reducing, capping, and stabilizing agents, adjusting the pH of water extracts under ultrasound conditions to form composites with silver nanoparticles. The resulting composites showed good anticancer effects and stability, with an IC50 of 0.25 μg/mL, demonstrating higher anticancer activity compared to herbal extracts from Piper longum, Crotalaria sessiliflora, and silver nanoparticle composites [73]. Selecting the most suitable stabilizer for drug characteristics is crucial to ensure the strong stability of silver nanoparticle composites with *Taxus* extracts. Kajani et al. mixed Ag NPs with KBr and then composited them with *Taxus* extracts. The resulting composite treated Caov-4 cells at 5 μg/mL for 72 h, causing cell death rates to reach 98% and above through apoptosis effects [74]. Pastorino synthesized nanotechnology polymer capsules using hydrochloride allylamine and sulfonylbenzene compound polyelectrolytes and MnCO_3_, injecting *Taxus* extracts into capsules for oral delivery purposes, ensuring smooth passage through the intestinal barrier and direct targeting of tumor sites, maximizing drug activity release. The capsules exhibited strong inhibitory activity against MCF-7 cells [75]. Further modifications to nanotechnology polymer capsules, such as using magnetic fields and compound composites for specific site recognition, can enhance drug release efficiency [76].

### 3.2. Liquid Dosage Forms

Liquid formulations mainly include decoctions, injections, tinctures, etc. Traditional Chinese medicine decoctions, a traditional preparation method in Chinese medicine, have a long history of using *Taxus* decoctions for treatment. Initially, decoctions were mainly used for rheumatic pain, edema, etc. With the advancement of modern medicine, the pharmacological effects and formulation exploration of *Taxus* liquid formulations have progressed further. Qu et al. prepared water decoctions from *Taxus* branches and leaves and conducted anticancer activity tests on human pancreatic cancer cells and pancreatic cancer nude mice. Immunohistochemical analysis showed that the expression level and intensity of tumor proliferation-related protein Ki-67 were reduced, and the inflammatory response was significantly weakened, indicating that *Taxus* water decoctions exhibit specific inhibitory capabilities against pancreatic cancer [77].

Paclitaxel injections, as the main direction of drug development from *Taxus* extracts and a common drug for clinical cancer treatment, are limited by paclitaxel’s insolubility in water, often requiring the use of Cremophor^®^ EL as a carrier for drug preparation. However, due to the toxicity of Cremophor^®^ EL itself, there are restrictions on the clinical application of paclitaxel injections. Wang et al. prepared paclitaxel nanoparticles via high-pressure homogenization with ethanol and ethyl acetate as the oil phase, mixed with Poloxamer 188 and PEG-400 aqueous surfactants, followed by high-pressure homogenization and lyophilization, to prepare a water-soluble paclitaxel suspension. The suspension showed better pharmacokinetics in the intravenous mouse model, with a larger area under the curve (AUC) 0–∞ (20.343 ± 9.119 μg·h/mL vs. 5.196 ± 1.426 μg·h/mL) and a higher clearance rate compared to paclitaxel solution (2.050 ± 0.616 L/kg·h vs. 0.556 ± 0.190 L/kg·h), indicating the prepared water-soluble paclitaxel suspension had good bioavailability [78]. Additionally, studies have shown that *Taxus* essential oil exhibits certain antibacterial activities against both Gram-negative and Gram-positive bacteria [79,80], and a Chinese patent CN103285082A disclosed a drug composition for treating gynecological inflammation, combining *Taxus* extracts and essential oil with cocoa butter, semi-synthetic fatty acid glycerides, etc. The prepared suspension increases the drug action area due to its molecular characteristics, facilitating absorption, using a direct administration route, and with strong practicability.

Vignolini et al. used *Taxus* leaves for active substance extraction and solid-phase extraction to obtain a *Taxus* tincture, which was rich in taxane compound content and had a free radical scavenging activity of 83.8% [81]. Tinctures have been used in China for a long time, with high concentrations of effective components and good preservative effects [82]. As an effective development path for traditional Chinese medicine, in-depth research should be conducted on this aspect of the formulation development of *Taxus*.

### 3.3. Semi-Solid Dosage Forms

Semisolid formulations primarily include topical preparations such as hydrogels. Nanocapsulation of compounds in drug delivery systems is a technological method that can modulate molecules’ physicochemical and pharmacological characteristics during transport. The preparation of drugs using colloidal systems can promote improved drug solubility, stability, bioavailability, and therapeutic effects [83]. A three-dimensional network system composed of hydrophilic polymers in hydrogels endows them with high biocompatibility. Hence, more studies are applying them in drug delivery systems [84]. He et al. used egg lecithin phosphatidylcholine, pluronic F68, and Cremophor^®^ EL as surfactants, with alcohol as a co-surfactant, to composite *Taxus* extracts. The resulting microemulsion showed a lower elimination rate in vivo compared to pure extracts, and rat in vivo experiments also indicated lower toxicity upon entering the body, with longer circulation times, and higher treatment efficiency [85].

Furthermore, a Chinese patent CN102895269B disclosed a *Taxus* and alginic mud composition with anti-eczema effects and its preparation method. It used a sodium alginate-potato starch-acrylic acid ternary reaction system for polymerization, and then the polymer was mixed with an alcohol extract obtained from *Taxus*. As a local medication, the prepared *Taxus* alginate gel exhibited inhibitory activity against Gram-negative and Gram-positive bacteria. In vivo experiments on eczema mouse models showed that the *Taxus* alginate gel could regulate inflammation factor levels, indicating its potential for clinical applications in anti-inflammatory and antibacterial treatments.

## 4. Discussion and Conclusions

Due to their significant pharmacological actions, the bioactive compounds in Taxus have become a valuable resource in modern drug development. With the continuous progress of medical science, effectively extracting and utilizing the medicinal components of Taxus and developing them into appropriate pharmaceutical dosage forms have become an important research hotspot in the scientific field. In our discussion on the development of various extraction methods and formulation techniques for Taxus, we have summarized the most critical methods and their applicability in Figure 1. This figure illustrates the various technologies and approaches utilized in the preparation and formulation of Taxus extracts, highlighting the connections between different extraction methods and their impact on formulation effectiveness and efficiency.

In the preparation methods of *Taxus* extracts, efficiently extracting active components from herbal materials is key. *Taxus* is rich in taxane compounds, flavonoids, polysaccharides, steroids, and alkaloids, with diverse extraction methods aimed at enhancing extraction efficiency and cost-effectiveness while considering the environmental sustainability impact. Various extraction technologies include Supercritical CO_2_ Extraction, Solvent Extraction, Ultrasound Extraction, MAE, Solid-Phase Extraction, PEF Extraction, and Enzymatic Extraction. Each method has its unique advantages and limitations. Supercritical CO_2_ extraction is especially suitable for extracting taxane compounds due to its environmental friendliness and efficiency. An efficient and selective extraction of specific active components can be achieved by optimizing extraction conditions, such as temperature, pressure, and co-solvent ratio. Solvent extraction is the most traditional method, where effective extraction of target components can be realized by selecting suitable solvents and optimizing extraction conditions. However, it may involve longer extraction times and more solvent consumption, needing improvement in extract selectivity. Solid-Phase Extraction significantly improves the selectivity of the extract and further enriches the extract. Ultrasound and MAE methods, as emerging extraction technologies, effectively break down plant cell walls in a short time through thermal and mechanical actions produced by ultrasounds or microwaves. These methods are characterized by simple operation and high extraction efficiency, which makes them especially suitable for the rapid extraction of temperature-sensitive compounds. PEF Extraction and Enzymatic Extraction technologies have also developed rapidly in recent years, applied in *Taxus* active component extraction to improve extraction efficiency and product yield. However, most of these methods involve organic solvents, with high environmental pressures and related costs.

Choosing the appropriate extraction method depends not only on the nature and efficiency of the target component but also on cost, environmental impact, and operational convenience. Therefore, comprehensive analysis and comparison of the characteristics and applicability of different extraction methods are crucial for optimizing the extraction process of *Taxus* components, improving the quality and yield of extracts. Currently, extraction is often a fusion of multiple techniques, combining strengths to significantly enhance extraction efficiency, making multidisciplinary crossover very important as an innovative approach. Recently, enzymatic catalysis technology has also developed rapidly, with its green and low-temperature features suitable for the extraction and separation of active components in *Taxus*, significantly improving extraction efficiency. Artificial intelligence technology is also making significant advances, with some statistics and algorithms greatly enhancing extraction efficiency, notably Response Surface Methodology (RSM), to enhance the efficiency and precision of extracting valuable compounds from plant materials. For example, RSM has been successfully utilized to optimize the extraction of taxanes from *Taxus* spp., significantly enhancing the purity of these valuable anticancer compounds [27,34,40,42,43,46]. It’s believed that integrating these technologies will further improve the extraction efficiency and selectivity of *Taxus* active products. In the future, with technological advancements and increased demand for environmentally friendly extraction methods, more efficient and green composite extraction technologies are expected to be developed and applied.

In traditional Chinese medicine formulation development, further research and application of *Taxus* extracts are crucial for improving therapeutic efficiency and drug utilization rates. Traditional Chinese medicine, especially *Taxus*, is a valuable resource in China’s medical field, but its traditional processing methods, such as decoction, have disadvantages like significant destruction of effective chemical components, low extraction rate of insoluble components, large dosage, poor uniformity, and low therapeutic efficiency. With the advancement of active component separation and extraction technology, the pharmacological activity of *Taxus* extracts has been extensively studied, and its active monomers have been gradually isolated and identified, promoting the process of developing extracts into different types of drug formulations. Currently, literature from both domestic and international sources has reported the application of *Taxus* extracts and its active components in various drug forms, including injections, oral capsules, and enteric-coated tablets, as well as enhancing their stability and bioavailability in solid, semi-solid, and liquid forms through specific nanotechnology. Particularly in solid formulations, such as *Taxus* extract complexes enhanced by nanotechnology, significant anticancer effects and antibacterial activities have been shown, providing new ideas for oral dosage form development. In liquid formulations, the application of *Taxus* decoctions in anticancer, antibacterial, and other fields has demonstrated the potential combination of traditional Chinese medicine preparation methods with modern medical research. The development of semi-solid formulations, such as *Taxus* hydrogels and alginic mud compositions, has shown advantages in improving drug solubility, stability, and therapeutic effects. The application of certain nanotechnologies and specific excipients significantly enhances the bioavailability of *Taxus* formulations, addressing the issue of low oral bioavailability. The development of some topical formulations has lowered the threshold for clinical application of *Taxus* extracts, expanding new clinical application potentials. These research outcomes not only promote the development of the Taxus industry but also provide important references and directions for the innovative development of pharmaceutical formulations worldwide, including traditional Chinese medicine.

However, *Taxus* drug development remains mainly at the research level, with relatively single and less innovative elements in dosage forms, many of which lack industrial characteristics and have not received clinical application approval, leaving substantial room for development in specific drug design and implementation. New technologies and excipients are needed to optimize drug formulations further, potentially improving the bioavailability of active components in *Taxus* extracts, enhancing their stability, and thus enhancing their therapeutic effects. In terms of formulations, the introduction of multidisciplinary crossover technologies or disruptive innovative technologies is needed to promote the modern development of *Taxus* drug formulations.

## Figures and Tables

**Figure 1 molecules-29-02291-f001:**
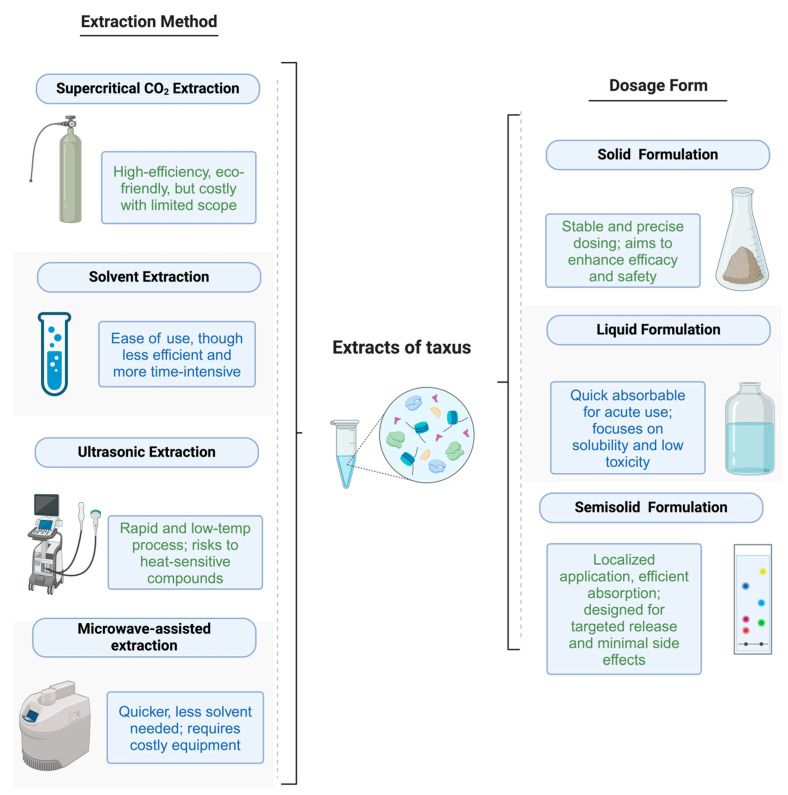
Preparation and Formulation Techniques of *Taxus*.

**Table 1 molecules-29-02291-t001:** Optimization of Supercritical CO_2_ Extraction Conditions for Target Compounds from *Taxus*.

Extracted Compound	Particle Size (Mesh)	Material-Solvent Ratio	Extraction Pressure	Extraction Temperature	Extraction Time	Co-Solvent	Efficiency/Yield	Reference
Paclitaxel	100	-	35 MPa	35 °C	120 min	-	96.10%	[12]
Paclitaxel	80	-	30 MPa	50 °C	120 min	Ethanol (10–15% water) r	93%	[13]
Paclitaxel	200	-	25 MPa	35 °C	-	-	-	[14]
10-DAB	50	1:1	27 MPa	35 °C	150 min	Anhydrous ethanol	0.55%	[15]
Flavonoids	-	-	22.6 MPa	46.4 °C	-	Ethanol (80.7%)	3.35–28 mg/g	[16]
Diterpenoids	-	-	10–35 MPa	40–70 °C	70–420 min-	-	35 mg/g	[17]
Volatile Oil	50	-	27 MPa	35 °C	150 min	Anhydrous ethanol	23.42%	[15]
Volatile Oil	20	-	25 MPa	45 °C	120 min	-	2.61%	[18]

**Table 2 molecules-29-02291-t002:** Optimization of different solvent for Target Compounds from *Taxus*.

Solvent	Extraction Compound	Solvent-Material Ration	Extraction Time	Yield	Reference
Water	Polysaccharide	-	12 h	54.3%	[26]
Ethanol	Total Flavonoid	32:1	86 min	128.1 mg/g	[27]
DCM	Taxane	2:1	-	0.028% from dry weight of bark, 0.008% from dry weight of leaves	[28]
Methanol	10-DAB	15:1	3 h	0.009%	[29]
Methanol: dichloromethane (1:1)	Paclitaxel	10:1	3 h	0.025%	[29]
Menthol:isopropanol (1:1)	Taxane	30:1	30 min	total extraction efficiency is 1.44 to the water extraction	[30]

**Table 3 molecules-29-02291-t003:** Summary of Ultrasound-Assisted Extraction Parameters and Yields for Various Taxus.

Taxus Species	Solvent and Concentration	Material-Liquid Ratio	Extraction Time	Ultrasound Temperature	Ultrasound Power	Additional Parameters	Extraction Yield	Reference
*Taxus wallichiana* var. *mairei*	50% Ethanol	1:15 (g/mL)	23 min	40 °C	-	-	1.18% of 10-DAB	[40]
*Taxus cuspidata*	-	53.23 mL/g	1.11 h	-	207.88 W	-	130.576 µg/g of paclitaxel	[41]
*Taxus x media*	Methanol with Ionic Liquids	1:10.5 (solid:liquid)	30 min	-	-	1.2% Ionic Liquid	-	[42]
*Taxus mairei*	66% Ethanol	1:31 (g/mL)	57 min	-	-	Two ultrasound extractions	106.58 mg/g of total flavonoids	[43]
*Taxus mairei*	60% Ethanol	14:1 (mL:g)	Enzyme: 50 min,	40 °C	120 W	0.15 g/L enzyme concentration	3.84% of proanthocyanidins	[44]
*Taxus chinensis*	19% Ethanol (pH 3)	12:1 (mL:g)	Enzyme: 52 min,	50 °C	150 W	0.10 g/L enzyme concentration	0.126% total alkaloids	[48]

**Table 4 molecules-29-02291-t004:** Comparison of different extraction methods of *Taxus* extracts.

Extraction Method	Advantages	Disadvantages	Extracted Compounds from *Taxus*	References
Supercritical CO_2_ Extraction	Efficient, eco-friendly, safe.	High cost, limited scope.	Taxane compounds, Alkaloids, Diterpenoids Flavonoids Volatile Oil	[9,10,11,65]
Solvent Extraction	Simple, low equipment need.	Low efficiency, time-consuming, solvent use.	Polysaccharides, Total Flavonoids, Alkaloids, Terpenes, Phenols	[21,22,23,24,25]
Water Extraction	Eco-friendly, simple.	Limited efficiency/purity for water-insoluble components.	Polysaccharides	[22,23]
Accelerated Solvent Extraction	Fast, improves recovery.	Risk of compound degradation, high equipment need.	Taxane compounds	[26,34]
Anti-solvent Recrystallization	Purifies specific components.	Complex, precise control needed.	Taxane compounds	[35,36]
Ultrasound Extraction	Simple, efficient, saves time/solvent.	May damage thermosensitive components.	Taxane compounds, Flavonoids, Alkaloids	[37,38,39,40,41,42,43,44]
Microwave-assisted extraction	Reduces time, lower solvent use.	Special equipment, high cost.	Taxane compounds, Polysaccharides	[49,50,51,52]
Solid-Phase Extraction Technology	Efficient in separating/enriching, improves yield.	Time-consuming, large solvent use, cumbersome.	10-DAB, 7-xyl-10-DAT, Paclitaxel, etc.	[57,58,59]
High-Intensity Pulsed Electric Field Extraction	Maintains bioactive component integrity.	Specialized equipment, precise control needed.	Taxane compounds	[60,61,62,63]
Enzymatic Assisted Extraction	Eco-friendly, suitable for low temperatures.	Cumbersome, high enzyme selection/use requirements.	Taxane compounds, Polysaccharides, etc.	[47,64]

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
