# Peer review of "Research Progress on Taxus Extraction and Formulation Preparation Technologies"

_molecules, 2024, doi:10.3390/molecules29102291_

Round 1

Reviewer 1 Report

Comments and Suggestions for Authors

Abstract

Taxus, as an evergreen tree spread throughout the world, contains a large number of bioactive components that play a crucial role in the pharmaceutical field. Taxus extracts, defined as a set of one or more bioactive compounds extracted from the genus Taxus spp., have become an important focus of modern cancer treatment research. This review article aims to delve into the scientific background of Taxus extracts and their considerable value in pharmaceutical research. It meticulously reviews and compares various advanced extraction techniques, such as supercritical extraction, ultrasonic extraction, microwave-assisted extraction, solid phase extraction, high-pressure pulsed electric field extraction, and enzymatic extraction, and They evaluate the advantages and limitations of each technology in aspects such as extraction efficiency, economic cost, operating time and environmental impact, with comprehensive analysis results presented in table form. In the field of drug formulation design, this article systematically analyzes the development strategies of solid, liquid and semi-solid dosage forms based on the unique physicochemical properties of Taxus extracts, their intended medical uses and their specific release characteristics, delving into the selection of excipients and the critical technical issues of the drug preparation process.

L:22 In all the extraction methods presented, in addition to the yields, the degree of purity with which the extract is obtained should be reported.

L:166 Table 1 is not referenced before it is presented

L:168 Solvent Extraction Method; it would be appropriate to include a table with the yields obtained in the extraction of taxus with each solvent.

L:294 Will acoustic impedance have any effect on taxus extraction?

L:385 Was there any relationship between microwave power of 215 W and the temperature generated in the extraction system?

Table 2.

I suggest including references

References

Because it is a review article I consider that at least 50% of the citations should be from the last 5 years, however they are only about 32% (considering 2019).

Author Response

Dear Reviewer,

Thank you for giving us an opportunity to revise our manuscript, we appreciate editor very much for their positive and constructive comments and suggestions on our manuscript entitled “Research Progress on the Preparation and Formulation Design Techniques of Medicinal Taxus Extracts” (molecules-2969079).

According to the reviewers’ comments, we have made necessary modifications to our manuscript and added additional explanations to make our results convincing. The revised paragraphs (sentences) are labeled in different colors.

We would like to express our great appreciation to you and reviewers for comments on our paper. Looking forward to hearing from you.

Thank you and best regards.

Sincerely,

Xinyu Gao

Weidong Xie

Reviewer 2 Report

Comments and Suggestions for Authors

In the attached manuscript, the authors provide an overview of methods that can be used for obtaining extracts from the genus Taxus. They describe methods that have been used to date, offering a detailed description of each technique and highlighting the advantages and disadvantages of each described method. Additionally, they outline how the obtained extract can be formulated. This review is interesting and provides a wealth of information. Below, I offer suggestions that could help the authors create an even better publication

1. In the reference list, several citations are provided not in English but in Chinese. It is assumed that the article aims to achieve global visibility, and for this reason, I request the authors to make the necessary adjustments

2. When discussing the genus Taxus, it would be beneficial to specify several specific species.

3. Supercritical fluid extraction is a method that has been known for several decades, and it is not considered an innovative method. Supercritical fluid extraction (SFE), a method extensively utilized in various industries such as food, pharmaceutical, and cosmetics for decades, is widely recognized for its efficiency in extracting compounds. The technology and equipment for SFE are well-established, with numerous standardized protocols in use, indicating its maturity rather than innovation.

4. In the chapter on Supercritical Fluid Extraction, the authors provide information on this technique, followed by a description of common solvent extraction procedures. The chapter title does not follow the document structure. I request the authors to organize this section much more clearly and concisely.

5. Generally, the document should have a much better structure in terms of the flow of ideas and data.

6. In the work where the authors discuss Ultrasonic-Assisted Extraction, they describe procedures based on the application of enzymes, with the enzyme technology being described later. It would have been better if the comparison of these two technologies and the results obtained were presented by the authors after explaining both methods.

7. In Chapter 2.8, titled "Novel Statistical and Algorithm Assisted Extraction," the authors briefly describe the application of statistical methods and Design of Experiments (DoE). The use of statistical tools is a part of the methodology in scientific work and is utilized across all mentioned techniques. There is no need to emphasize this separately unless the authors wish to delve deeper into the methodology related to this area.

8. Table 2 should be made simpler and more organized.

9. In the chapter that covers solid dosage forms, the authors mention topical preparations, which fall under semi-solid dosage forms.

10. In the manuscript, a figure or diagram is inserted without a title, nor is there any connection made between it and the rest of the text.

Comments on the Quality of English Language

"I have no comments regarding the English language

Author Response

(The authors gave the same response as above.)

Reviewer 3 Report

Comments and Suggestions for Authors

In this review article, the authors have systematically discusses the development strategies for solid, liquid,  and semi-solid dosage forms based on the unique physicochemical properties of Taxus extracts, their  intended medical uses, and specific release characteristics, delving deeply into the selection of excipients and the critical technical issues in the drug preparation process. The article is generally organized and well written, only few concerns need to be addressed:

-The title"Research Progress on the Preparation and Formulation Design Techniques of Medicinal Taxus Extracts" is too long and I suggest to shorten it.

-The content of Table 1 need to be more uniform.

- More details need to be included in section 2.8.

- The quality of the graphical abstract in page 17 need to be improved and please make the text more concise.

Comments on the Quality of English Language

Minor editing.

Author Response

(The authors gave the same response as above.)

Round 2

Reviewer 2 Report

Comments and Suggestions for Authors

I take this opportunity to thank the authors for accepting most of the suggestions. At this time, I have no further comments and suggest that the paper be accepted.

Author Response

Thank you very much for taking the time to review our manuscript and for your valuable comments. We appreciate your insights and the detailed feedback you provided, which will undoubtedly help us improve the quality of our work. Thank you once again for your thoughtful and constructive critique.